# Integrating Transcriptomics and Metabolomics to Explore the Novel Pathway of *Fusobacterium nucleatum* Invading Colon Cancer Cells

**DOI:** 10.3390/pathogens12020201

**Published:** 2023-01-28

**Authors:** Xinyu Wu, Jinzhao Xu, Xiaoying Yang, Danping Wang, Xiaoxi Xu

**Affiliations:** 1Key Laboratory of Dairy Science, Ministry of Education, Northeast Agricultural University, Harbin 150030, China; 2College of Food Science, Northeast Agricultural University, Harbin 150030, China

**Keywords:** *Fusobacterium nucleatum*, extracellular vesicles, colon cancer cells, transcriptomics, metabolomics

## Abstract

Colorectal cancer (CRC) is a malignancy with a very high incidence and mortality rate worldwide. *Fusobacterium nucleatum* bacteria and their metabolites play a role in inducing and promoting CRC; however, no studies on the exchange of information between *Fusobacterium nucleatum* extracellular vesicles (*Fn*evs) and CRC cells have been reported. Our research shows that *Fusobacterium nucleatum* ATCC25586 secretes extracellular vesicles carrying active substances from parental bacteria which are endocytosed by colon cancer cells. Moreover, *Fn*evs promote the proliferation, migration, and invasion of CRC cells and inhibit apoptosis; they also improve the ability of CRC cells to resist oxidative stress and SOD enzyme activity. The genes differentially expressed after transcriptome sequencing are mostly involved in the positive regulation of tumor cell proliferation. After detecting differential metabolites using liquid chromatography–tandem mass spectrometry, *Fn*evs were found to promote cell proliferation by regulating amino acid biosynthesis in CRC cells and metabolic pathways such as central carbon metabolism, protein digestion, and uptake in cancer. In summary, this study not only found new evidence of the synergistic effect of pathogenic bacteria and colon cancer tumor cells, but also provides a new direction for the early diagnosis and targeted treatment of colon cancer.

## 1. Introduction

Colorectal cancer (CRC) is one of the most common malignancies worldwide, with top three incidence (10%) and mortality (9.4%) rates according to updated data from the International Cancer Center GLOBOCAN database [1]. With new CRC cases and deaths expected to increase to 3.2 million and 1.6 million, respectively, by 2040, there is an urgent need to better understand and act on the characteristics of CRC. The causative factors and pathogenesis of CRC are not only related to human life and dietary habits; it is well known that intestinal epithelial cells are constantly exposed to millions of microbial communities, also known as intestinal flora, which are strongly associated with intestinal health [2]. Some of these microorganisms play a probiotic role in protecting intestinal health, such as *Lactobacillus plantarum* [3], *Lactobacillus rhamnosus GG* [4], *Bifidobacteria* [5], etc. Some microorganisms play an important pathogenic role in the initiation or promotion of intestinal diseases, such as *Salmonella* [6], *Helicobacter pylori* [7], *Escherichia coli* [8], etc. One recent study suggested that bidirectional communication between microbes and host cells in the intestinal ecosystem may not involve direct cellular contact [9]. Bacterially secreted extracellular vesicles serve as a mechanism of communication between bacteria as well as between bacteria and host cells, enabling the long-distance delivery of active compounds to bacteria in a protected environment, avoiding direct contact between bacteria and host cells to achieve information exchange with host cells. Additionally, some strains that establish a symbiotic relationship with their hosts release extracellular vesicles as a mechanism to deliver active bacterial substances that regulate important functions of the host cells [10].

Bacterial extracellular vesicles (BEVs) are spherical lipid bilayer nanostructures ranging in size from 20 to 300 nm. They are produced by Gram-negative and Gram-positive bacteria and are natural carriers of bioactive substances secreted by bacteria including lipopolysaccharides (LPS), peptidoglycans, lipids, proteins, nucleic acids, toxins, virulence factors, and small molecules [11]. It has been shown that the extracellular vesicles of certain probiotics have a modulating effect on intestinal immunity [12]; for example, EVs of *Akkermansia muciniphila* can modulate the inflammatory response and reduce lipopolysaccharide-induced intestinal barrier damage [13,14], and EVs of *Lactobacillus rhamnosus* GG and *Lactobacillus plantarum* Q7 may alleviate DSS-induced ulcerative colitis by modulating the intestinal microbiota [15,16]. In contrast, EVs of certain Gram-negative pathogenic bacteria can cause the bacteria to attack host cells and disrupt host mechanisms [17]; for example, EVs of *Pseudomonas aeruginosa* cause a cellular inflammatory response [18], and EVs of *H. pylori* promote the formation of a biofilm and strengthen its structural integrity, thereby enhancing bacterial pathogenicity [19].

*Fusobacterium nucleatum* (*Fn*) is a specialized anaerobic Gram-negative pathogenic bacterium which is one of the most abundant commensal bacteria in the oral cavity. As a conditional pathogen, it can cause oral and extraoral inflammatory diseases [20] and adverse pregnancy outcomes [21]. Recent studies have pointed to its close association with the development of CRC in humans [22], The abundance of *Fusobacterium nucleatum* was significantly higher in colorectal cancer tissues compared with normal tumor-adjacent tissues [23]. Studies have shown that *Fusobacterium nucleatum* activates the TLR4/NF-kB signaling pathway [24,25] and upregulates the expression of inflammatory factors including IL-1β, IL-6, IL-17F, and TNF-α to promote inflammatory diseases [26]. Its secreted amyloid mucin FadA enhances bacterial virulence and promotes biofilm formation [27]. It can bind to E-cadherin in colorectal cancer cells and upregulate Annexin a1 to activate the Wnt/β-catenin signaling pathway, promoting the proliferation of colon cancer cells [28]. Its outer membrane secretes lectin Fap2 that mediates binding to Gal-GalNAc, which, in turn, is overexpressed in CRC tumors, promoting cell proliferation [28]. It also binds to TIGIT receptors on natural killer (NK) cells and tumor-infiltrating lymphoid (T) cells, inhibiting the activity of immune cells and enabling tumor cell escape [29]. It is still unknown whether *Fusobacterium nucleatum* carries multiple pathogenic factors through secreted extracellular vesicles, which are recognized and endocytosed by tumor cells such as CRC, facilitating the bacterial infection of cells to perform information interactions with tumor cells. *Fusobacterium nucleatum* may also regulate the activity of CRC cells through this mechanism. Therefore, we integrated metabolomic and transcriptomic analyses to determine the mechanism by which *Fusobacterium nucleatum* extracellular vesicles invade colon cancer cells, which can provide new ideas for the early diagnosis, prevention, and treatment of colorectal cancer.

## 2. Materials and Methods

### 2.1. Preparation of Fusobacterium nucleatum ATCC25586

*Fusobacterium nucleatum* ATCC25586 (*Fn*) was purchased from Beijing Beina Chuanglian Institute of Biotechnology. The strain was inoculated in thioglycolate liquid medium and incubated anaerobically at 37 °C (80% N_2_, 10% CO_2_, 10% H_2_). The growth cycle of *Fn* was determined by Bioscreen C and the structure of the bacterium was observed by scanning electron microscopy and transmission electron microscopy.

### 2.2. Isolation of Fnevs

The supernatant obtained after the centrifugation of *Fn* at 3000× *g* for 10 min at 4 °C was filtered through a 0.45 μm filter to remove cell debris. The filtrate was transferred to a new centrifuge tube with exosome concentration solution (ESC) reagent at 4 °C for 20 h. *Fn*evs were precipitated by centrifugation at 10,000× *g* for 1 h. After the precipitated *Fn*evs had been resuspended in PBS, the supernatant was taken as the crude extracted *Fn*evs after centrifugation at 12,000× *g* for 2 min. Finally, the crude extracted *Fn*evs were transferred to an exosome purification filter (EPF) column at 4 °C and centrifuged at 3000× *g* for 10 min to finally obtain the purified *Fn*evs, which were stored at −80 °C, following the manufacturer’s instructions for the Umibio Exosome Extraction Kit. To assess the concentration of purified *Fn*evs, the protein content of Fnevs was determined by measuring the absorbance of the samples at 562 nm using a BCA kit (Dalian Meilun Bio, Dailan, China) with an enzyme-labeled instrument.

### 2.3. Transmission Electron Microscopy (TEM)

To examine the morphology of *Fn*evs using transmission electron microscopy (Hitachi, Japan), 10 µL of purified *Fn*evs was fixed with 2.5% glutaraldehyde and placed on a 300-mesh copper grid. Grids were stained with 2% uranyl acetate.

### 2.4. Scanning Electron Microscopy (SEM)

The SEM sample preparation method for *Fn*evs is basically the same as that for bacteria: 20 μL of purified Fnevs was pipetted onto φ12 mm L-polylysine-coated glass crawlers and left at room temperature until they were about to dry completely, followed by the same procedure as for the preparation of bacterial SEM.

### 2.5. Nanoparticle Tracking Analysis

*Fn*evs were removed from the 10 μL mixture and diluted to 30 μL. The particle size distribution curve and concentration of *Fn*evs were measured using a NanoFCM N30E nanoflow detector.

### 2.6. Protein Identification and Analysis of Fnevs

The proteins were extracted by adding appropriate amounts of SDT (4% (*w*/*v*) SDS, 100 mM Tris/HCl, pH 7.6) lysate to the samples; then, the proteins were quantified using the BCA method. For each sample, 20 µg of protein was added to the appropriate amount of 5X loading buffer, heated in a boiling water bath for 5 min, SDS-PAGE electrophoresis was performed (4–20% pre-made gradient gel, constant pressure 180 V, 45min), and Kemas Brilliant Blue R-250 staining was conducted. Each sample was trypsinized using the filter-aided proteome preparation (FASP) method [30]. The peptide was desalted using a C18 Cartridge, and the peptide was lyophilized and re-solubilized by adding 40 μL of 0.1% formic acid solution; then, the peptide was quantified (OD280). Separation was then performed using NanoElute, an HPLC liquid phase system with a nanoliter flow rate. Buffer A was 0.1% formic acid aqueous solution; buffer B was 0.1% formic acid acetonitrile aqueous solution (99.9% acetonitrile). The chromatographic column was equilibrated with 95% of liquid A. The samples were separated on a C18 reverse-phase analytical column (Thermo Scientific EASY column, 25 cm, ID75 μm, 1.9 μm) with a flow rate of 300 nL/min. The samples were separated by chromatography and analyzed by mass spectrometry using a timsTOF Pro mass spectrometer operating in positive ion mode at 1.5 kV. Both MS and MSMS were analyzed by TOF in the mass range of *m*/*z* 100–1700. The data acquisition mode was run in parallel accumulated sequence fragmentation (PASEF) mode with specific parameters: ion mobility coefficients (1/K0) of 0.6 to 1.6 Vscm2. MaxQuant software [31] was used for library identification and quantitative analysis, and relevant bioinformatics analysis was performed.

### 2.7. Cell Source and Culture

Colorectal cancer cell lines (Caco-2 cells and HT-29 cells) were purchased from Shanghai Cell Bank. Caco-2 and HT-29 cells were isolated from human colorectal cancer primary tumor tissue. The cells were cultured in DMEM (Melun, Dalian, China) supplemented with 10% fetal bovine serum (FBS), 1% penicillin and streptomycin (Beyotime, Shanghai, China) at 37 °C with 5% CO_2_ in an incubator (ThermoFisher, Shanghai, China).

### 2.8. Fluorescent Labeling Fnevs

To identify the internalization of *Fn*evs uptake by colorectal cancer cell lines, *Fn*evs were first labeled using the exosomal fluorescent labeling dye PKH26 (Uimibio, Shanghai, China). The dye working solution was added to centrifuge tubes containing *Fn*evs, vortex-shaken and mixed, and then incubated for 10 min at rest and protected from light. Subsequently, 10 mL of 1X PBS was added to the incubated exosome–dye complex and mixed well. The exosomes were extracted again according to the exosome extraction method to remove excess dye. Then, 200 μL of 1X PBS was used to resuspend the precipitate, which was the stained exosomes. Colon cancer cells were incubated with labeled *Fn*evs in an incubator at 37 °C with 5% CO_2_ for 24 h. Nuclei were stained with DAPI and visualized under a laser confocal microscope TCS SP8 (Leica, Wetzlar, Germany).

### 2.9. Detection of Cell Proliferation by CCK-8

To detect cell proliferation using CCK-8, 96-well plates were inoculated at a density of 1 × 10^4^ cells per well, and 100 μL of medium was added to each well. After 12 h of overnight incubation, the cells were treated with different concentrations of *Fn*evs (25 μg/mL, 50 μg/mL, and 100 μg/mL) or PBS, and compared with the live bacteria treatment group (MOI of 100:1). Co-incubation was performed for 24 h and 48 h at 37 °C with 5% CO_2_. Then, following the guidance in the CCK-8 kit (Meilun, Shaoxing, China), the absorbance at 450 nm was measured with an enzyme-labeled instrument.

### 2.10. Flow Cytometry Analysis of Apoptosis

The Annexin V-FITC kit (Biyuntian Biotechnology Co., Ltd., Shanghai, China) was used to detect the degree of apoptosis. Cells were inoculated in 6-well plates and treated with PBS or 50 μg/mL *Fn*evs for 48 h at 37 °C with 5% CO_2_. Then, the cells were digested with EDTA-free trypsin and washed twice with PBS. The cells were collected and centrifuged at 1000× *g* for 5 min. The supernatant was discarded, 200 µL of Annexin V-FITC conjugate was added to gently resuspend the cells, 5 µL of Annexin V-FITC was added, the cells were incubated for 15 min at room temperature and protected from light; 5 µL of PI staining was added 5 min before loading, 300 µL of Annexin V-FITC conjugate was added, and apoptosis was detected using an Accuri C6 plus flow cytometer (Becton Dickinson, Franklin Lakes, NJ, USA).

### 2.11. Detection of Intracellular ROS Level and SOD Enzyme Activity

Intracellular ROS levels were detected using a reactive oxygen species assay kit (Beyotime, Shanghai, China); a reactive oxygen species control (Rosup) was used as a stimulus. Colon cancer cells were inoculated in 6-well plates and stimulated with reactive oxygen species (Rosup), then treated with PBS or 50 μg/mL *Fn*evs at 37 °C with 5% CO_2_ for 24 h, followed by fluorescence detection.

A superoxide dismutase (SOD) kit (Nanjing Jiancheng, Nanjing, China) was used to detect the viability level of SOD in cells. Colon cancer cells were inoculated in 6-well plates and treated with PBS or 50 μg/mL *Fn*evs at 37 °C with 5% CO_2_ for 24 h; then, cells were collected to detect intracellular SOD enzyme activity levels.

### 2.12. Wound-Healing Assay and Transwell Assay for Cell Migration and Invasion

Cells were resuspended with 200 μL of serum-free DMEM and added to the top chamber of a Transwell with an 8.0 μm pore membrane (Corning, NY, USA), and DMEM containing 10% FBS was added to the lower chamber. For cell invasion experiments, the upper chamber pore membrane needed to be pre-coated with Matrigel (Corning, NY, USA) then treated with PBS or 50 μg/mL *Fn*evs. After co-culture at 37 °C in 5% CO_2_ for 24 h, cells in the lower chamber were fixed with 4% paraformaldehyde. After washing twice with PBS, the bottom of the chamber was stained with 0.1% crystalline violet (Biyuntian China) for 10 min. Stained cells were observed with an inverted light microscope (Leica, Germany) and five fields of view were randomly selected for counting.

Wound healing assays were used to determine the ability of cells to migrate. After growth to 80% confluence in 6-well plates, wounds were created in the cells by scraping off the monolayer using a plastic pipette tip. Cells were then washed three times with PBS to remove debris, treated with PBS or 50 μg/mL *Fn*evs, and incubated with serum-free medium at 37 °C with 5% CO_2_ for 24 h. The cells were then photographed.

### 2.13. Transcriptome Sequencing (RNA-Seq) and Analysis

HT-29 cells were treated with PBS or 50 μg/mL *Fn*evs at 37 °C with 5% CO_2_ for 24 h. The medium was discarded, 3 mL of PBS buffer was added to the cell culture flask, and 1 mL of TRIZOL was added for every 5 × 10^6^ cells to extract total RNA. The A260/A280 absorbance ratio of RNA samples was measured using a Nanodrop ND-2000 (Thermo Scientific, Waltham, MA, USA) and the RIN value of RNA was determined using an Agilent Bioanalyzer 4150 (Agilent Technologies, Santa Clara, CA, USA). Only quality-checked RNA can be used for library construction. PE libraries were prepared according to the instructions of the ABclonal mRNA-seq Lib Prep Kit (ABclonal, Wuhan, China). The mRNA was purified from 1 μg of total RNA using oligo (dT) beads, and then fragmented in ABclonal First Strand Synthesis Reaction Buffer. Subsequently, the first strand of cDNA was synthesized using random primers and reverse transcriptase (RNase H) using the mRNA fragment as a template, followed by the second strand of cDNA using DNA polymerase I, RNAseH, buffer, and dNTPs. The synthesized double-stranded cDNA fragments were ligated with splice sequences and used for PCR amplification. PCR products were purified and the library quality was assessed using an Agilent Bioanalyzer 4150. Finally, sequencing was performed using the NovaSeq 6000 (or MGISEQ-T7) sequencing platform PE150 read length. The data generated from the Illumina platform were used for bioinformatics analysis, and all the analyses were performed by Shang Hai APPLIED PROTEIN TECHNOLOGY.

### 2.14. LC-MS/MS for Differential Metabolite Analysis

HT-29 cells were treated with PBS or 50 μg/mL *Fn*evs at 37 °C in 5% CO_2_ for 24 h. After removing the medium, the cells were washed 2–3 times with PBS buffer and the cell samples were collected with a cell spatula, and an appropriate amount of sample was added to a pre-chilled methanol/acetonitrile/water solution (2:2:1, *v*/*v*/*v*), mixed by vortexing, sonicated at low temperature for 30 min, left at −20 °C for 10 min, and centrifuged at 14,000× *g*, 4 °C. For mass spectrometry, 100 μL of acetonitrile aqueous solution (acetonitrile:water = 1:1, *v*/*v*/*v*) was added, vortexed, and centrifuged at 14,000× *g* for 15 min at 4 °C. The supernatant was taken from the sample for analysis. The sample was separated on an Agilent 1290 Infinity LC HILIC column with the following conditions: column temperature, 25 °C; flow rate, 0.5 mL/min; injection volume, 2 μL. Mobile phase composition A was water + 25 mM ammonium acetate + 25 mM ammonia; Mobile phase composition B was acetonitrile. The gradient elution procedure was as follows: 0~0.5 min, 95% B; 0.5~7 min, B varied linearly from 95% to 65%; 7–8 min, B varied linearly from 65% to 40%; 8~9 min, B was maintained at 40%; 9~9.1 min, B varied linearly from 40% to 95%; 9.1~12 min, B was maintained at 95%; throughout the analysis, the sample was placed at 4 min, B varied linearly from 95% to 95%; 9.1~12 min, B varied linearly from 95% to 95%; B varied linearly from 65% to 40%; 8~9 min, B was maintained at 40%; 9~9.1 min, B was linearly changed from 40% to 95%; 9.1~12 min, B was maintained at 95%. The sample was placed in an autosampler at 4 °C during the whole analysis. Subsequent mass spectrometric analysis was performed with a Triple TOF6600 mass spectrometer (AB SCIEX) in electrospray ionization (ESI) positive and negative ion modes, respectively. The detected metabolites were subjected to structural identification, and the data were pre-processed to complete the quality evaluation and bioinformatics analysis.

### 2.15. Statistical Analysis

All experiments were repeated at least three times, and the data were analyzed and graphed using GraphPad Prism 8.0. All data were expressed as the mean ± standard deviation (SD). Differences between the control group and *Fn*evs treatment were analyzed by *t*-test. *p* < 0.05 was considered a statistically significant difference.

## 3. Results

### 3.1. Growth Kinetic Curve and Structural Observation of Fusobacterium nucleatum ATCC25586

The growth curves of bacteria were measured using a Bioscreen C growth curve analyzer, and the logarithmic growth period of *Fn* was determined. The different concentrations of the bacterial solution had different absorbance values at a wavelength of 600 nm; thus, the absorbance value reflected the logarithmic growth period and stable period of *Fn*. The absorbance of the bacterial solution was measured at an interval of 30 min (six replicate wells were set up at the same time) to obtain the *Fn* growth curve. As shown in Figure 1a, the logarithmic growth phase began at 6 h and the stable growth phase began at 27 h. After performing scanning electron microscopy, it could be observed that the two ends of the *Fn* bacterium exhibited a typical shuttle structure, and adhered to the surface of the glass crawl sheet to form a dense biofilm structure, as shown in Figure 1b. Therefore, the magnification was enhanced to observe the surface of the bacterium; a single, tiny vesicle body was shed from the surface of the bacterium, so it is inferred that the strain can secrete extracellular vesicles. The internal structure of *Fn* observed in combination with transmission electron microscopy revealed a relatively clear edge of the bacterium with an intact cell membrane, a homogeneous solid body in the center, and a more homogeneous extracellular protein layer around it (Figure 1c).

### 3.2. Characterization of Extracellular Vesicles of Fusobacterium nucleatum ATCC25586 (Fnevs)

*Fn*evs in the supernatant of *Fusobacterium nucleatum* ATCC25586 were isolated by ultracentrifugation, and their uniformly spherical microstructures were observed by scanning electron microscopy (Figure 1d). Furthermore, *Fn*evs with the signature teato-like lipid bilayer structure of extracellular vesicles were observed by transmission microscopy (Figure 1e). The extracted Fnevs were removed from 10 μL and diluted to 30 μL and analyzed using a nanoparticle tracer; the average diameter of Fnevs particles was determined to be approximately 177.1 nm with a concentration of 2.1 × 10^10^ particles/mL (Figure 1f). The protein concentrations of *Fn*evs were determined using a BCA kit at 0.205 μg/μL, and the extracted proteins were detected by SDS-PAGE electrophoresis, as shown in Figure 1g.

### 3.3. Protein Identification of Fnevs

In total, 579 proteins were detected in *Fn*evs, and the identified proteins were subjected to subcellular localization prediction and classification statistics; the results are shown in Figure 1h. Among the proteins identified, 486 cytoplasmic proteins (83.9%) accounted for the highest percentage; there were 83 outer membrane proteins (14.3%), 77 extracellular proteins (13.2%), and 63 extracellular and other proteins. The proteins identified by Go functional analysis (Figure 1I) revealed that 81.8% of them were involved in the regulation of cellular processes and metabolic processes in bacteria. The remaining proteins were involved in the biosynthesis and regulation of nucleic acids and the transport of nutrients such as carbohydrates, lipids, and amino acids; some were involved in stress response, antioxidant response, and bacterial adhesion function. These findings suggest that *Fn*evs contain some of the more bioactive proteins of the parent bacteria.

### 3.4. Internalization of Fnevs

According to the laser confocal images, it could be observed that fluorescently labeled *Fn*evs were absorbed by Caco-2 cells and HT29 cells into the cytoplasm and accumulated around the blue nucleus to emit red fluorescence (Figure 2a). However, it was also apparent that more labeled *Fn*evs were observed in HT-29 cells compared with Caco-2 cells, probably due to the variability between different cells, whereas HT-29 cells were more sensitive to *Fn*evs.

### 3.5. Effect of Fnevs on the Proliferation and Apoptosis of Colon Cancer Cells

To verify the effect of *Fn*evs on colon cancer cells, we used the CCK-8 method to detect the viability of Caco-2 cells and HT29 cells after treatment with different concentrations of *Fn*evs. As shown in Figure 2b for Caco-2 cells, the effect of *Fn*evs on cell proliferation increased with concentration at 24 h, but was less different from that of the live bacteria treatment group, whereas at 48 h, the cell proliferation rate was significantly higher in the treated group above 50 μg/mL (34.45%) than in the live bacteria group (9.55%). For HT-29 cells, the promotion of cell proliferation by *Fn*evs was concentration-dependent compared with the control group within 24 h. The cell viability was significantly increased and significantly more than twofold higher than that of the live bacterium-treated group after treatment with 50 μg/mL of *Fn*evs; the cell proliferation rate was as high as 61.51% in the group treated with 100 μg/mL.

We then examined the effect of *Fn*evs on the apoptosis of Caco-2 cells and HT29 cells by Annexin V/PI double staining. The sum of Q2 and Q4 regions represented the total apoptosis rate as detected by flow cytometry. Compared with the control group, for Caco-2 cells, the apoptosis rate was reduced by 55.25% after *Fn*evs treatment; for HT-29 cells, the apoptosis rate was reduced by 38.1% after *Fn*evs treatment; *Fn*evs treatment also reduced the apoptosis rate of colorectal cancer cells (Figure 2c).

### 3.6. Effect of Fnevs on the Migration, Invasion, and Healing Ability of Colon Cancer Cells

The effect of *Fn*evs on the migration and invasion of colon cancer cells was detected using the Transwell method; migrating and invading cells were stained with crystal violet. The results are shown in Figure 2d. The level of cell migration and invasion was increased in the group treated with the addition of *Fn*evs compared with the control group. Therefore, it is believed that *Fn*evs can promote the migration and invasion of colon cancer cells. After ImageJ was employed to randomly select different fields of view for counting and averaging, it was found that the migration and invasion rates of Caco-2 cells treated with *Fn*evs increased by 56.86% and 35.54%, respectively, and those of HT-29 cells increased by 64.19% and 44.97%, respectively, compared with the control group. Therefore, it is believed that *Fn*evs can promote the migration and invasion of colon cancer cells.

Through the cell scratch assay, we found that *Fn*evs enhanced the healing ability of colon cancer cells, as shown in Figure 2e. The width of scratches in the *Fn*evs treatment group was much narrower than that in the control group. By measuring the width between two scratches, the healing ability of Caco-2 cells and HT-29 cells was increased by 38% and 44%, respectively, after *Fn*evs treatment.

### 3.7. Effect of Fnevs on the ROS Level and SOD Enzyme Activity in Colon Cancer Cells

The proliferation and metastasis of tumor cells can be inhibited by increasing the concentration of ROS in tumor cells, resulting in oxidative stress, and thus, apoptosis. The test stimulated a sharp increase in ROS levels in tumor cells by the positive control (Rosup), while the ROS levels in the cells with added *Fn*evs were reduced. Caco-2 cells decreased by 39.44% and HT-29 cells decreased by 44.72%, indicating that *Fn*evs can reduce the oxidative stress response induced by external conditions and maintain the normal physiological activity of colon cancer cells; the fluorescence results are shown in Figure 2f.

Superoxide dismutase (SOD) is an important active enzyme in cells, and the physiological activity of cells can be determined by measuring the activity of this enzyme. For Caco-2 cells, the *Fn*evs treatment increased the enzyme activity by an average of 16.23 (U/mgprot), and for HT-29 cells, the *Fn*evs treatment increased the enzyme activity by 67.45 (U/mgprot). *Fn*evs were tested to promote the activity of the SOD enzyme in colon cancer cells (Figure 2g) and enhance the ability of colon cancer cells to fight free radicals and protect cancer cells from oxidative damage, thus enhancing the activity of colon cancer cells.

### 3.8. Differential Gene Expression and Enrichment Pathways in Fnevs-Treated Colon Cancer Cells

To further determine the mechanism of *Fn*evs invasion of colon cancer cells, we performed transcriptome sequencing and bioinformatics analysis of differentially expressed genes in HT29 cells after *Fn*evs treatment. The 4508 upregulated mRNAs and 3146 downregulated mRNAs were screened, as shown in the volcano plot (Figure 3a) and heatmap (Figure 3b). To explore the function of differentially expressed genes, we performed Go enrichment analysis (Figure 3c) to show the functional annotation of the top ten most significantly associated differential genes and to describe the properties of genes and gene products in organisms in three aspects: biological process (BP), molecular function (MF), and cellular component (CC). We found that differentially expressed mRNAs are involved in positive regulation of the cell proliferation process. Upregulated genes such as DHRS2, ID2 [32], CCNA1 [33], EGR2 [34], WNT4 [35], RIPK4 [36], PDK4 [37], IL-11 [38], and MAPK4 play a role in promoting the development of colon cancer and various malignancies [39]. IL-16 plays a role in promoting cellular inflammation [40]; BCl-6 plays a role in inhibiting apoptosis in colon cancer cells [41]; and NFKBIZ plays a role in promoting the expression of the IL-17 inflammatory signaling pathway [42]. Downregulated genes, such as DMTN, play a role in inhibiting the proliferation and metastasis of colon cancer tumors [43]. gLP2R and CERS2 play a role in inhibiting intestinal inflammation [44,45]. The KEGG pathway enriched by these genes is shown in Figure 3d. In the KEGG pathway maps with a high number of differential genes, such as Herpes simplex virus 1 infection, transcriptional misregulation in cancer, and the cGMP-PKG signaling pathway, we observed downregulation of the P53 gene with a log2FoldChange value of −2.4826. It is known that P53 is an important oncogene and plays an important role in the P53 signaling pathway to inhibit tumor development; downregulation of the P53 gene in *Fn*evs-treated colon cancer cells promoted their proliferation, migration, and invasion ability, while inhibiting the expression of the downstream apoptosis-promoting Bax gene. We also found that upregulation of the ERG gene in the pathway, with a log2FoldChange value of 6.6051, promoted the expression of downstream target genes, such as MMP, including MMP9, which was involved in promoting cell migration and invasion, which may explain the increased migration and invasion rates of *Fn*evs-treated colon cancer cells. Additionally, it was found that a gene named CCL2, which was not expressed in the blank group but was expressed in the *Fn*evs-treated group and is now considered to be a novel oncogene, was downregulated by TLR3 expression in the KEGG pathway. We found that the inhibition of downstream target gene IRAK1 expression could promote Ikk expression and lead to IkB phosphorylation and NFkB dissociation, and finally promote CCL2 expression. By analyzing these differentially expressed genes and their functions, we have gained a deeper understanding of the role of *Fn*evs in promoting colon cancer cell activity.

### 3.9. Differential Metabolites and Metabolic Pathways in Fnevs-Treated Colon Cancer Cells

The univariate analysis for metabolites detected in positive and negative ion mode is shown in the form of a volcano plot, as shown in Figure 4a. The validity of the model was ensured by multidimensional statistical analyses such as principal component analysis (PCA) and orthogonal partial least squares analysis (OPLS-DA), as well as permutation tests (Figure 4b). In total, 429 cationic metabolites and 272 anionic metabolites were screened by combining the databases. OPLS-DA VIP > 1 and *p* value < 0.05 were used as the screening criteria for significantly different metabolites, among which 58 significantly different metabolites were obtained in positive ion mode, 20 of which were significantly upregulated and 38 were significantly downregulated; 20 significant difference metabolites were obtained in negative ion mode, 6 of which were significantly upregulated and 14 of which were significantly downregulated. Changes in the number of significantly different metabolites identified are visualized in bar graphs in Figure 4c. This experiment was further analyzed by clustering these significantly different metabolites; the results are presented in the form of a hierarchical clustering heatmap, as shown in Figure 4d. Metabolites clustered in the same cluster had similar expression patterns and may have similar functions or share the same metabolic processes or cellular pathways. In order to visualize the significance of the differences between the metabolic pathways and the enrichment of metabolites, this experiment combined different metabolites in the two groups of samples in positive and negative ion mode, and enrichment analysis was performed through the KEGG pathway. The top 16 metabolic pathways with the highest significance were selected based on *p* values; the results of the enrichment analysis are presented in the form of bubble plots (Figure 4e), and the top five metabolic pathways with the most significant differences are presented in the form of cluster heatmaps (Figure 4f).

Significantly upregulated metabolites, such as phenylalanine and proline, which are essential amino acids, have been suggested to exhibit significantly higher phenylalanine levels in colon cancer tumor tissue compared with normal epithelial mucosal tissue [46], indicating that phenylalanine may be an amino acid which plays a key role in colon cancer proliferation and metastasis. Proline can be used for the synthesis of cytoplasmic proteins to promote cell proliferation by means of NADP(H) produced by mitochondria [47]. The differential expression fold change (fold change) of furanophylline was 16.62, which is an inhibitor of cytochrome P450 [48]. It is known that CYP450 can cause an increase in reactive oxygen species, leading to mitochondrial dysfunction and promoting apoptosis, whereas the upregulation of furanophylline can effectively inhibit CYP450 and thus ensure the activity of colon cancer cells, which could also explain why colon cancer cells proliferated after *Fn*evs treatment. This may also explain why the oxidative stress capacity of *Fn*evs-treated colon cancer cells was increased. N-palmitoyl-d-synuclein, a ceramide, regulates cellular activity and promotes the development of colon cancer by acting through the Wnt/β-catenin signaling pathway [49,50]. Oxypurinol is a metabolite with a high differential ploidy, with a differential expression ploidy (fold change) value as high as 16.45. It has the function of reducing lipid peroxide accumulation and scavenging the production of cellular superoxide radicals; some studies have found that its expression level is significantly increased in gastric tumors [51,52]. The increase in SOD enzyme activity in colon cancer cells after *Fn*evs treatment may be related to this. Metabolites that were significantly downregulated were mostly amino acids such as tryptophan, histidine, lysine, argininosuccinate, tyrosine, and O-acetyl-1-serine. Studies have shown that colon cancer cells require more tryptophan than normal cells to convert to kynurenine for their own proliferation and metastasis, and that decreased plasma tryptophan levels correlate with CRC tumor development [53]. Reduced serum histidine levels are associated with intestinal barrier damage, ulcerative colitis, and various cancer diseases [54]. At the same time, the downregulation of pyruvate helps colon cancer cells to avoid cell death due to HDAC1/HDAC3 inhibition [55]. A decrease in the metabolic level of argininosuccinic acid, an intermediate in the tricarboxylic acid cycle, demonstrates the enhanced glycolytic demand of colon cancer cells [56]. These differential metabolites and their enriched metabolic pathways are important biomarkers and pathways of action for *Fn*evs to invade colon cancer cells.

## 4. Discussion

In the tumor microenvironment, extracellular vesicles of microbial origin play an important role in disease diagnosis, monitoring, and treatment [57]. *Fusobacterium nucleatum* (*Fn*) plays an important role in the development of colon cancer as a conditionally pathogenic bacterium. It may be symbiotic with colon cancer cells, and with the concept of “intracellular bacteria” [58], we are more curious about the mechanism of communication between *Fn* and colon cancer. However, most existing studies have focused on the relationship between live bacteria and metabolites of *Fusobacterium nucleatum* and colon cancer and other malignancies, although the association between its secreted extracellular vesicles and colon cancer tumor cells is still unknown. Therefore, we identified its characterized structure by extracting the extracellular vesicles of *Fusobacterium nucleatum* and determining the size and morphology of the extracellular vesicles secreted by *Fusobacterium nucleatum*. The active substance can better achieve the invasion of host cells.

After further exploration, we found that extracellular vesicles (*Fn*evs) secreted by *Fusobacterium nucleatum* could be recognized and taken up by colon cancer cells. For strains that establish a symbiotic relationship with their hosts, they release BEVs as a mechanism to deliver bacterial mediators that regulate important physiological functions of the host. It is believed that the uptake of BEVs by mammalian host cells is mainly achieved by endocytosis. For example, phagocytosis is the main pathway for the internalization of BEVs by phagocytic cells of the immune system (e.g., neutrophils, macrophages, and dendritic cells). In contrast, in non-phagocytic cells, such as epithelial cells, the pathways that allow BEVs to enter the cell include macrocytosis, lattice-protein-mediated endocytosis, and lipid-raft-mediated endocytosis, which may be dependent on caveolin [59,60]. These pathways may sometimes occur together, but the exact mode of uptake depends on their size and the active substance they carry. Therefore, more accurate determination of the de-activating substances carried by them from the parental bacteria is required. However, after performing the protein identification of *Fn*evs, we believe that they carry active protein components from the parental bacteria. It has been suggested that the secretion mechanism of extracellular vesicles in Gram-negative pathogenic bacteria is mainly due to the “blistering” effect of the outer membrane, which was confirmed by electron microscopy. Through in vitro co-culture, we found that *Fn*evs exhibited strong effects in promoting colon cancer cell activity, including promoting colon cancer cell proliferation and inhibiting apoptosis, enhancing the oxidative stress capacity of colon cancer cells, reducing abnormal ROS levels caused by external stimuli, and promoting cellular SOD enzyme activity. Colon cancer cells infected with *Fn*evs exhibited enhanced migration, invasion, and healing abilities. Through transcriptomic and metabolomic assays, we found that differentially expressed genes are mostly involved in the positive regulation of tumor cell proliferation, and further analysis of differential metabolites also clarified the signature substances produced after the *Fn*evs treatment of colon cancer cells. However, considering the possible inaccuracy of single histological analysis, we combined the results of transcriptomics and metabolomics to search for genes that exhibit key changes after the *Fn*evs infection of colon cancer cells and the metabolites they affect; in integrating the enriched pathways of transcriptomics and metabolomics, we found 72 KEGG pathways in which differentially expressed genes and differentially expressed metabolites were jointly involved. We identified one pathway with a high enrichment factor as “Central carbon metabolism in cancer”. In this pathway, we found that the upregulation of the SLC1A5 gene, which encodes a glutamine transporter protein on the cell membrane surface, promoted the production of 2-Oxoglutarate from glutamine by colon cancer cells to provide a constant source of energy for cell proliferation via the TCA cycle, whereas the involvement of glutamate in proline metabolism in mitochondria promoted the significant upregulation of proline as a metabolite. Activation of the cell membrane surface tyrosine kinase receptor RTKS promotes RAS expression which, in turn, leads to phosphorylation of the downstream genes MEK and ERK, and finally leads to phosphorylation of C-Myc in the nucleus and promotes the expression of the SLC1A5 gene. This not only provides a new direction for our subsequent studies, but also provides predictive information on targets for the prevention and treatment of the proliferation of colon cancer tumors caused by Fnevs, which should explored in more depth.

## 5. Conclusions

In conclusion, our results show that *Fusobacterium nucleatum* secretes extracellular vesicles (*Fn*evs) carrying active components from the parental bacteria, a novel model which enables them to modulate host cells without contacting them, and the ability of *Fn*evs to promote various aspects of colon cancer cell activity was examined. The mechanism of *Fn*evs’ action was predicted by examining the differentially expressed genes and the metabolites of Fnevs after infection with colon cancer cells. This discovery not only expands the pathways of action of pathogenic microorganisms in the gut on host cells, but also provides new ideas for the early diagnosis and prevention of colon cancer caused by *Fusobacterium nucleatum*, a pathogenic bacterium. This mechanism should be explored further.

## Figures and Tables

**Figure 1 pathogens-12-00201-f001:**
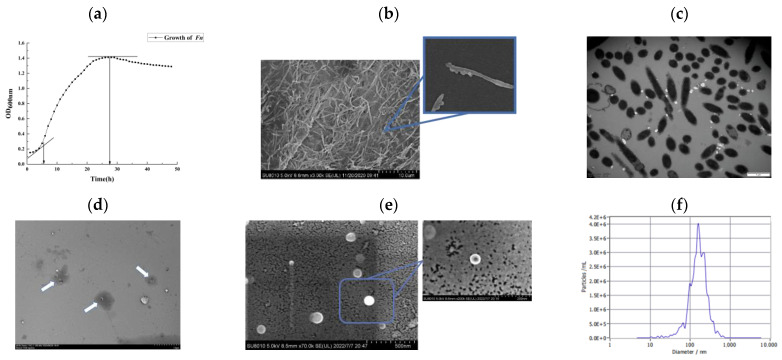
Characterization of *Fn* and *Fn*evs. (**a**) *Fn* growth curve. (**b**) Scanning electron microscopy (SEM) images of *Fn* magnified. (**c**) Transmission electron microscopy (TEM) images of *Fn.* (**d**) Transmission electron microscopy (TEM) images of *Fn*evs. Scale bar: 100 nm per small grid. (**e**) Scanning electron microscopy (SEM) images of *Fn*evs. (**f**) Nanoparticle tracing analysis of *Fn*evs. (**g**) SDS-PAGE of *Fn*evs (M: marker; 1: *Fn*evs). (**h**) Subcellular localization pie chart. (**i**) Go functional analysis of *Fn*evs protein.

**Figure 2 pathogens-12-00201-f002:**
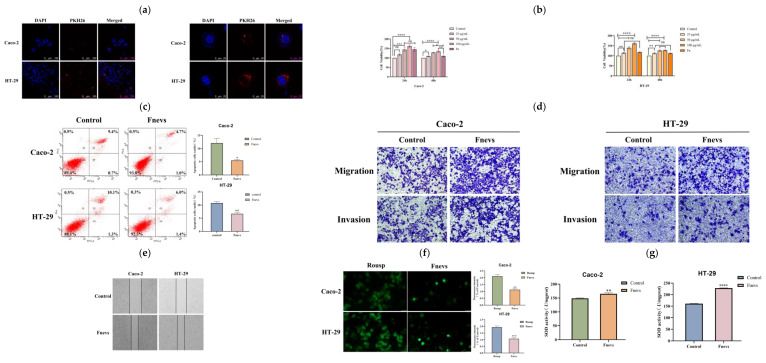
Effect of *Fn*evs on colon cancer cells. (**a**) Fluorescence images of *Fn*evs taken up by colorectal cancer cells. Scale bar: 100 μm and 25 μm. (**b**) The viability of Caco-2 and HT-29 cells after being treated with *Fn* and *Fn*evs, compared with the control group. (**c**) The cells were stained with Annexin V/PI, and the number of apoptotic cells was determined by flow cytometry; the difference is shown in a bar chart. (**d**) *Fn*evs promoted the migration and invasion of Caco-2 and HT-29 cells. Scale bar: 100 μm. (**e**) Scratch healing ability of cells after *Fn*evs treatment. (**f**) Effect of *Fn*evs on the fluorescence level of cellular ROS after stimulation. Scale bar: 50 μm. (**g**) Changes in Caco-2 and HT-29 cells of SOD enzyme activity after *Fn*evs treatment. All experiments were repeated at least three times (* *p* < 0.05, ** *p* < 0.01, *** *p* < 0.001, **** *p* < 0.0001).

**Figure 3 pathogens-12-00201-f003:**
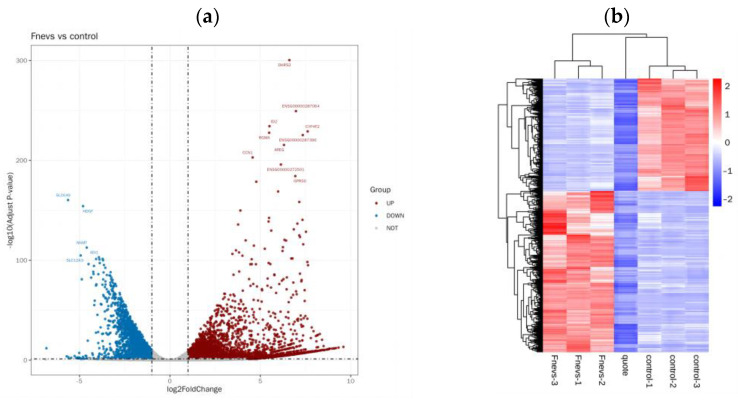
Transcriptome sequencing after the *Fn*evs infection of colon cancer cells. (**a**) Volcano plots. (**b**) Heatmaps of differentially expressed genes between the control group and *Fn*evs group. (**c**) GO enrichment analysis of differentially expressed genes. (**d**) Differentially expressed genes in the KEGG pathway enrichment analysis.

**Figure 4 pathogens-12-00201-f004:**
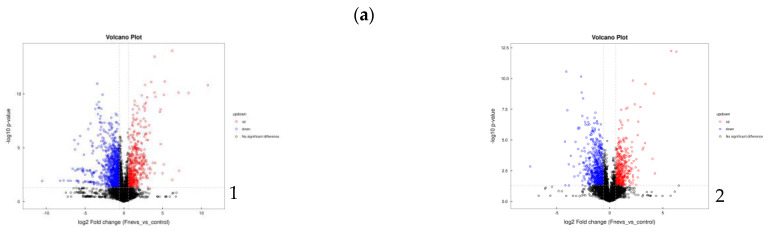
Differential metabolites after *Fn*evs treatment of colon cancer cells. (**a**) Volcano plots of univariate statistical analysis (1, positive ion mode; 2, negative ion mode). (**b**) Multivariate statistical analysis (PCA, OPLS-DA, permutation test) (1, positive ion mode; 2, negative ion mode). (**c**) Differential metabolites fold change analysis column chart (1, positive ion mode; 2, negative ion mode). (**d**) Heatmaps of differential metabolites hierarchical clustering analysis. Red represents relatively high expression, blue represents relatively low expression, and metabolites with close expression patterns are clustered under the same cluster on the left (1, positive ion mode; 2, negative ion mode). (**e**) Enrichment KEGG pathway air bubble diagram: the darker the color and the larger the bubble, the more important the pathway. (**f**) Heatmaps of the top five enrichment pathways (1, biosynthesis of amino acids; 2, central carbon metabolism in cancer; 3, protein digestion and absorption; 4, aminoacyl-tRNA biosynthesis; 5, 2-oxocarboxylic acid metabolism).

## Data Availability

The original contributions presented in the study are included in the articlel, and further inquiries can be directed to the corresponding authors.

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
