# Peer review of "Integrating Transcriptomics and Metabolomics to Explore the Novel Pathway of Fusobacterium nucleatum Invading Colon Cancer Cells"

_pathogens, 2023, doi:10.3390/pathogens12020201_

Round 1

Reviewer 1 Report

This study aimed to explore the impact of Fusobacterium nucleatum extracellular vesicles (Fnevs) on colorectal cancer cells. Previous research has shown that Fusobacterium nucleatum is over-expressed in colon cancer, and multiple mechanisms through which it may contribute to the development of cancer have been identified. However, little is known about the effects of Fnevs on cancer cells. This study represents a novel contribution to the field and is of significant interest.

Despite this, I have several concerns about the study design and the approaches used in this work.

1. One concern with the study design is that it only evaluated the in vitro effects of extracellular vesicles from a single Fusobacterium nucleatum strain on two colon cancer cell lines. This means that the results may not be generalizable. There was no in vivo work or independent validation of the findings, which limits the significance of the results.

2. The results of the study are largely descriptive. While the figures may provide some information about the effects of Fnevs on cancer cells, they do not provide details about the numerical values, magnitudes, or significance of the changes observed. It would be helpful if the main text of the study elaborated more about these aspects of the results. In addition, the figures could be improved with better resolution to make them more interpretable.

3. Fusobacterium nucleatum is a facultative anaerobe, and the growth of this bacterium in the study was measured under anaerobic incubation conditions. Nevertheless the co-culture conditions were not described, including the duration, oxygen condition, and protocol which may have impacted the findings. This is important as large parts of the works are based on in vitro co-culture conditions.

4. The transcriptomic findings, which relate to the expression of genes in the cancer cells, were not superficial in the main text and were not replicated using an independent method. This raises questions about the reliability and robustness of these findings. It would be important to validate these key findings using a different approach or at a different molecular level in order to increase confidence in the results.

5. The metabolite findings were largely descriptive. While some upregulated metabolites are mentioned, there was no follow-up work to investigate the effects of these differentially abundant metabolites. This means that it is not clear how these changes in metabolite levels may have impacted the cancer cells or the overall findings of the study. Further investigation of these metabolites could provide additional insights into the effects of Fnevs on cancer cells.

Author Response

Dear Reviewer:

Thank you for giving us an opportunity to revise our manuscript. We appreciate your constructive comments and suggestions on our manuscript (pathogens-2145659). We have modified the manuscript accordingly, and detailed point-by-point responses are listed below. And we use MDPI's language touch-up service, We hope that the revision will meet the standards of the journal and your expectations.

Response to Reviewer 1 Comments

Point 1:One concern with the study design is that it only evaluated the in vitro effects of extracellular vesicles from a single Fusobacterium nucleatum strain on two colon cancer cell lines. This means that the results may not be generalizable. There was no in vivo work or independent validation of the findings, which limits the significance of the results.

Response 1: Thank you very much for your reminders and valuable comments on our research design. We chose to use Fusobacterium nucleatum ATCC25586 because we felt it was reliable and credible as a standard strain isolated from the oral cavity and commonly used in studies of colon cancer and other tumors after referring to research articles summarizing the co-culture of bacteria with colon cancer and other tumor cells.And as a dominant pathogenic strain it plays an important role in causing symbiosis and disease development of many other pathogens. I have appended the study references for this strain below.Your suggestion is very beneficial to me, and follow up our group will screen more Fusobacterium nucleatum subspecies for In-depth researching.

  1. Fusobacterium nucleatumpromotes colorectal cancer cells adhesion to endothelial cells and facilitates extravasation and metastasis by inducing ALPK1/NF-κB/ICAM1 axis. Gut Microbes. 2022 Jan-Dec;14(1):2038852. doi: 10.1080/19490976.2022.2038852.
  2. Perturbed human sub-networks by Fusobacterium nucleatum candidate virulence proteins. Microbiome. 2017 Aug 10;5(1):89. doi: 10.1186/s40168-017-0307-1.
  3. ANGPTL4-Mediated Promotion of Glycolysis Facilitates the Colonization of Fusobacterium nucleatumin Colorectal Cancer. Cancer Res. 2021 Dec 15;81(24):6157-6170. doi: 10.1158/0008-5472.CAN-21-2273.
  4. The Pathogenic Effects of Fusobacterium nucleatumon the Proliferation, Osteogenic Differentiation, and Transcriptome of Osteoblasts. Front Cell Dev Biol. 2020 Sep 11;8:807. doi: 10.3389/fcell.2020.00807.
  5. Fusobacterium nucleatum Accelerates Atherosclerosis via Macrophage-Driven Aberrant Proinflammatory Response and Lipid Metabolism. Front Microbiol. 2022 Mar 11;13:798685. doi: 10.3389/fmicb.2022.798685.
  6. Genome sequence and analysis of the oral bacterium Fusobacterium nucleatum strain ATCC 25586. J Bacteriol. 2002 Apr;184(7):2005-18. doi: 10.1128/JB.184.7.2005-2018.2002.

Since our study is new and preliminary, and considering that in vivo experiments in mice are subject to environmental and individual variability, we chose two colon cancer cell lines with high accuracy and reproducibility in order to determine more accurately whether Fnevs can be endocytosed by colon cancer tumor cells from a molecular perspective. We have chosen two colon cancer cell lines in our study that are highly accurate and reproducible. It has also been shown that HT-29 cell line can form tumors by subcutaneous transplantation in mice. Our current study focuses on the characterization of Fusobacterium nucleatum secreted bacterial extracellular vesicles and the determination and investigation of their uptake by colon cancer cells, which we believe is a novel mechanism for Fusobacterium nucleatum to act on colon cancer cells and may not require direct contact between live bacteria and cells, as the secreted extracellular vesicles can carry the active protein material of the parental bacteria and be endocytosed by colon cancer cells. We have demonstrated that Fnevs can be endocytosed by colon cancer cells and promote their proliferation, migration invasion healing, slow down their apoptosis and enhance their healing ability through in vitro cell co-culture experiments. Moreover, we found that Fnevs can resist oxidative stimulation induced by external stimuli and increase the SOD enzyme activity of colon cancer cells. We aim to find the differentially expressed genes and metabolites of Fnevs treated colon cancer cells through transcriptomic and metabolomic techniques, and predict its possible pathways of action through the KEGG pathway for more reference of other researchers, and our group will continue to screen the significantly upregulated metabolites and differentially expressed genes for more in-depth study through in vivo validation experiments in mice. Our group will continue to screen for significantly upregulated metabolites and differentially expressed genes for more in-depth study through in vivo validation experiments.

Point 2:The results of the study are largely descriptive. While the figures may provide some information about the effects of Fnevs on cancer cells, they do not provide details about the numerical values, magnitudes, or significance of the changes observed. It would be helpful if the main text of the study elaborated more about these aspects of the results. In addition, the figures could be improved with better resolution to make them more interpretable.

Response 2: Thank you for your valuable comments!

  • In the results analysis section, I have corrected and red-colored the original descriptive report by adding specific values and multiples of change, and discussing the significance behind the values. Hopefully, this will give you a better visualization of the test results and meet your expectations.
  • We apologize for not providing clear figures in the initial manuscript submission due to my negligence, and again, I apologize for the bad experience with your review, This time I have re-uploaded the figures in my resubmitted revision and I have also attached my figures in the attached reply file to you for your re-review of my data. Thank you again for all your hard work.

Point 3:Fusobacterium nucleatum is a facultative anaerobe, and the growth of this bacterium in the study was measured under anaerobic incubation conditions. Nevertheless the co-culture conditions were not described, including the duration, oxygen condition, and protocol which may have impacted the findings. This is important as large parts of the works are based on in vitro co-culture conditions.

Response 3: Thank you for your valuable comments!

I apologize for not expressing clear co-culture conditions in the manuscript due to my negligence. I have added specific co-culture conditions in red font in the methodology of the new revised manuscript.

We set up the current experimental protocol by referring to the in vitro co-culture study of Fusobacterium nucleatum and colon cancer cells, and referred to the methods used in other in vitro co-culture experiments with extracellular vesicles of anaerobic bacteria and cells, I have attached the relevant references below. And giving a detailed description of the specific co-culture conditions for each experiment.

  • For 8. Fluorescent labeling Fnevs: Colon cancer cells were incubated with labeled Fnevs in an incubator at 37°C with 5% CO2 for 24 hours.
  • For 9. Detection of cell proliferation by CCK-8:The Colon cancer cells were treated with different concentrations of Fnevs (25 μg/mL, 50 μg/mL and 100 μg/mL) or PBS, and compared with the live bacteria treatment group(MOI of 100:1), Coincubation for 24 h and 48 h at 37°C with 5% CO2.
  • For 10. Flow cytometry analysis of apoptosis: Colon cancer cells were inoculated in 6-well plates and treated with PBS or 50 μg/mL Fnevs for 48 h at 37°C with 5% CO2.
  • For 11. Detection of intracellular ROS level and SOD enzyme activity:
  • ROS level: Colon cancer cells were inoculated in 6-well plates and stimulated with reactive oxygen species (Rosup) and then treated with PBS or 50 μg/mL Fnevs at 37 °C with 5% CO2 for 24 h.
  • SOD enzyme activity: Colon cancer cells were inoculated in 6-well plates and treated with PBS or 50 μg/mL Fnevs at 37°C with 5% CO2 for 24 h.
  • For12. Scratch test and Transwell assay for cell migration and invasion:
  • Wound-Healing Assay: Wound healing assays were used to determine the ability of cells to migrate. After growth to 80% confluence in 6-well plates, wounds were created in the cells by scraping off the monolayer using a plastic pipette tip. Cells were then washed three times with PBS to remove debris, treated with PBS or 50 μg/mL Fnevs, and incubated with serum-free medium at 37 °C with 5% CO2 for 24 h.The cells were then photographed.
  • Transwell assay for cell migration and invasion :Colon cancer cells were added to transwell plates chambered simultaneously with PBS or 50 μg/mL Fnevs and Co-culture at 37°C with 5% CO2 for 24 hours.
  • For13. Transcriptome Sequencing(RNA-seq)and analysis: HT-29 cells were treated with PBS or 50 μg/mL Fnevs at 37 °C with 5% CO2 for 24 h.
  • For14. LC-MS/MS for Differential metabolites analysis: HT-29 cells were treated with PBS or 50 μg/mL Fnevs at 37 °C with 5% CO2 for 24 h.

I have attached below the references of Fusobacterium nucleatum  and anaerobic extracellular vesicles co-cultured with cells, thank you again for your valuable suggestions so that I can better improve the quality of my manuscript.

  • Fusobacterium nucleatumpromotes colorectal cancer cells adhesion to endothelial cells and facilitates extravasation and metastasis by inducing ALPK1/NF-κB/ICAM1 axis. Gut Microbes. 2022 Jan-Dec;14(1):2038852. doi: 10.1080/19490976.2022.2038852.
  • The Pathogenic Effects of Fusobacterium nucleatumon the Proliferation, Osteogenic Differentiation, and Transcriptome of Osteoblasts. Front Cell Dev Biol. 2020 Sep 11;8:807. doi: 10.3389/fcell.2020.00807.
  • Liu L, Liang L, Yang C, Zhou Y, Chen Y. Extracellular vesicles of Fusobacterium nucleatum compromise intestinal barrier through targeting RIPK1-mediated cell death pathway. Gut Microbes. 2021 Jan-Dec;13(1):1-20. doi: 10.1080/19490976.2021.1902718.
  • Impact of Escherichia coliOuter Membrane Vesicles on Sperm Function. Pathogens. 2022 Jul 10;11(7):782. doi: 10.3390/pathogens11070782.
  • The Fruits of Paris polyphyllaInhibit Colorectal Cancer Cell Migration Induced by Fusobacterium nucleatum-Derived Extracellular Vesicles. Molecules. 2021 Jul 4;26(13):4081. doi: 10.3390/molecules26134081.

Point 4:The transcriptomic findings, which relate to the expression of genes in the cancer cells, were not superficial in the main text and were not replicated using an independent method. This raises questions about the reliability and robustness of these findings. It would be important to validate these key findings using a different approach or at a different molecular level in order to increase confidence in the results.

Response 4: Thank you for your valuable comments!

Thank you for your constructive comments, the main objective of our study was to determine whether Fusobacterium nucleatum as a pathogenic bacterium can secrete extracellular vesicles, and in our study we determined this doubt, and by characterizing and identifying it we also determined that it can carry protein components of the parental bacterium. And we demonstrated that it can be endocytosed by colon cancer cells while promoting its related activity, which is one of the highlights of our study. We found that Fusobacterium nucleatum can avoid direct contact between bacteria and cells through a new pathway of secreting extracellular vesicles, which can achieve the regulatory effect of pathogenic bacteria on colon cancer cells

We also agree with your opinion that the discussion of transcriptomics results is not deep enough, and that the expression of genes is not validated and has some inaccuracy, and we have thought about it. In designing the study methodology, after reviewing the relevant references, we found that the application of transcriptome detection of differential genes and their enriched KEGG pathways to achieve the prediction of probing the impact of target material tumor cells is credible to some extent, and I have attached the relevant references below.

  • Integrated transcriptome and in vitro analysis revealed anti-proliferative effect of citral in human stomach cancer through apoptosis. Sci Rep. 2019 Mar 19;9(1):4883. doi: 10.1038/s41598-019-41406-8.
  • Transcriptome profiling links the intrinsic properties of human prostate basal cells to prostate cancer aggressiveness. Mol Cell Oncol. 2016 Mar 30;3(3):e1168508. doi: 10.1080/23723556.2016.1168508.
  • The transcriptome difference between colorectal tumor and normal tissues revealed by single-cell sequencing. J Cancer. 2019 Oct 11;10(23):5883-5890. doi: 10.7150/jca.32267.

 Our group also hopes that our findings and predictions can provide more researchers with ideas and directions, and in the newly submitted manuscript, we have analyzed the differentially expressed genes in the transcriptome assay in more depth and summarized the upstream and downstream pathways and important genes involved in the regulation of the KEGG pathway, and compared them with the existing studies, and we have combined the differentially expressed genes and metabolites in the discussion section, which is also a joint analysis of transcriptomics and metabolomics, from the perspective of cause to effect. We hope that our corrections will meet your expectations and that our group will use your suggestions for in vivo validation and more in-depth studies to verify our predicted changes in subsequent studies.

Point 5:The metabolite findings were largely descriptive. While some upregulated metabolites are mentioned, there was no follow-up work to investigate the effects of these differentially abundant metabolites. This means that it is not clear how these changes in metabolite levels may have impacted the cancer cells or the overall findings of the study. Further investigation of these metabolites could provide additional insights into the effects of Fnevs on cancer cells.

Response 5: Thank you for your valuable comments!

We have thought deeply about your suggestion, and in our revised manuscript we have reflected the multiplicity of differential metabolites in visual numbers, and in the metabolomics analysis section the significantly up- and down-regulated metabolites are discussed more in relation to the existing studiesand, Moreover, by searching the relevant literature, we added and discussed in detail the regulatory effects of certain significantly up- or down-regulated metabolites on colon cancer cells, and more fully explained our experimental results including the effects of Fnevs on proliferation apoptosis, migration, invasion, as well as ROS levels and SOD enzyme activity of colon cancer cells. In recent years the application of metabolomics techniques to detect differential metabolites to predict biomarkers in the tumor environment has been more applied, and I have attached references below.

  • Metabolomics in cell culture--a strategy to study crucial metabolic pathways in cancer development and the response to treatment. Arch Biochem Biophys. 2014 Dec 15;564:100-9. doi: 10.1016/j.abb.2014.09.002.
  • Development of an Optimized Protocol for NMR Metabolomics Studies of Human Colon Cancer Cell Lines and First Insight from Testing of the Protocol Using DNA G-Quadruplex Ligands as Novel Anti-Cancer Drugs. Metabolites. 2016 Jan 15;6(1):4. doi: 10.3390/metabo6010004.
  • Cell metabolomics study on the anticancer effects of Ophiopogon japonicusagainst lung cancer cells using UHPLC/Q-TOF-MS analysis. Front Pharmacol. 2022 Sep 16;13:1017830. doi: 10.3389/fphar.2022.1017830.

By detecting the differential metabolites, we can clarify how the increase in colon cancer cell activity caused by Fnevs causes changes in colon cancer cells at the metabolic level, and we also want to make predictions by clarifying these differential metabolites as biomarkers formed after infection of colon cancer cells by Fnevs.We hope that our prediction can provide new ideas for research scholars. Meanwhile, our group will also start the in-depth study of specific metabolites acting on the tumor microenvironment in vivo according to your suggestion, thank you very much for your valuable comments.

Reviewer 2 Report

In this manuscript, Wu et al. attempt to integrate transcriptomics and metabolomics to explore the novel pathway of Fusobacterium nucleatum invading colon cancer cells.

The authors indeed provided an interesting observation that the Fusobacterium nucleatum ATCC25586 secreted extracellular vesicles (Fnevs) endocytosed by colon cancer cells and provided them with proliferative and survival advantages. Unfortunately, much of the data presented in this manuscript are of poor quality and on many occasions, it is hard to interpret authors' claim.

Besides, the main caveat of this manuscript is that it fails to propose and validate a novel molecular mechanism that explains how Fusobacterium nucleatum extracellular vesicles (Fnevs) promote Colorectal cancer (CRC). Rather it simply confirms previous work that, in some cases, is not cited or discussed. In addition, the manuscript seems to run in different directions.

Comments.

1.     All 4 figures provided in the manuscript were presented poorly by the authors, in most cases figure panel captions are misaligned or missing from the figure (See Figures 1B and 1C).

2.      Figure 2 of this manuscript discusses a very important aspect of this study; however I find it hard to interpret the author’s claim because of the poor quality of the data provided. Authors should provide high-resolution images for all confocal microscopy and FACS data.

3.     CCK8 assay alone will not be sufficient to measure the CRC proliferation rate, please supplement with an additional highly sensitive assay.

4.     The author’s interpretation of transcriptomics and metabolomics is highly subjective, authors should instead discuss the novel findings and provide few validations.

5.     Authors should carefully review the manuscript text to avoid typos and syntax error

 Few examples: Line 30 – ‘Databas’ instead ‘Database’. 

Author Response

Dear Reviewer:

Thank you for giving us an opportunity to revise our manuscript. We appreciate your constructive comments and suggestions on our manuscript (pathogens-2145659). We have modified the manuscript accordingly, and detailed point-by-point responses are listed below. And we use MDPI's language touch-up service, We hope that the revision will meet the standards of the journal and your expectations.

Response to Reviewer 2 Comments

Point 1:All 4 figures provided in the manuscript were presented poorly by the authors, in most cases figure panel captions are misaligned or missing from the figure (See Figures 1B and 1C).

Response 1: Thank you for your valuable comments! We apologize for not providing clear figures in the initial manuscript submission due to my negligence, and again, I apologize for the bad experience with your review, This time I have re-uploaded the figures in my resubmitted revision and I have also attached my figures in the attached reply file to you for your re-review of my data. Thank you again for all your hard work.

Point 2: Figure 2 of this manuscript discusses a very important aspect of this study; however I find it hard to interpret the author’s claim because of the poor quality of the data provided. Authors should provide high-resolution images for all confocal microscopy and FACS data.

Response 2:  Thank you for your valuable comments! We apologize for not providing clear figures in the initial manuscript submission due to my negligence, and again, I apologize for the bad experience with your review, This time I have re-uploaded the figures in my resubmitted revision and I have also attached figure-2 below to you for your re-review of my data. Thank you again for all your hard work.

Point 3:CCK8 assay alone will not be sufficient to measure the CRC proliferation rate, please supplement with an additional highly sensitive assay.

Response 3: Thank you for your valuable comments!

I supplement with an additional highly sensitive assay-EdU cell proliferation assay.

The principle of EdU detection of cell proliferation relies onEdU (5-ethynyl-2'-deoxyuridine), which is a novel thymidine (thymidine deoxyriboside, thymidine) analogue that can be used in the DNA synthesis Moreover, the ethynyl group on EdU can bind to the fluorescence-labeled small molecule azide probe and click reaction. Therefore, the newly synthesized DNA will fluoresce in red, while all the nuclei will fluoresce in blue by Hoechst33342 solution. Three different fields of view were taken for each test group, and the relative positive rate was analyzed according to ImageJ software.

  • Development of ethynyl-2'-deoxyuridine chemical probes for cell proliferation. Bioorg Med Chem. 2016 Sep 15;24(18):4272-4280. doi: 10.1016/j.bmc.2016.07.021.
  • The Raman way of following endothelial cell proliferation in vitro and ex vivo. Biosens Bioelectron. 2022 Nov 15;216:114624. doi: 10.1016/j.bios.2022.114624.

By comparison with the Control and Fn-treated groups(MOI=100:1), the 50 μg/mLFnevs-treated group showed higher relative red fluorescence intensity, more newly synthesized DNA and faster cell proliferation

Point 4:The author’s interpretation of transcriptomics and metabolomics is highly subjective, authors should instead discuss the novel findings and provide few validations.

Response 4: Thank you for your valuable comments! 

Since our study is a relatively new discovery, we focused on the ability of Fusobacterium nucleatum to secrete extracellular vesicles and to clarify by characterization and protein assay that it does carry the active component from the parental bacteria, while we demonstrated by in vitro co-culture experiments that Fnevs can be endocytosed by colon cancer cells and promote proliferation, migration, invasion and wound healing, enhance their oxidative stress capacity, reduce unexpected elevation of ROS levels and enhance cellular SOD enzyme activity. The discovery of a novel pathway for Fusobacterium nucleatum to act on colon cancer cells through the secretion of extracellular vesicles is a highlight of our study. We further analyzed the differential genetic and metabolic changes in colon cancer cells induced by Fnevs by histological means, and the analysis of the results of the above experiments can also explain the increased activity of colon cancer cells after treatment with Fnevs, and after a prediction of the differential genetic and metabolic effects of the KEGG pathway, it also provides a new direction for other researchers, and we group will also refer to your comments for multifaceted argumentation in later studies.

To the discussion, I added a joint analysis of transcriptomics and metabolomics.The integration of multi-omics data for analysis can compensate for the data deficiencies of single-omics data analysis, and the mutual validation between individual histology data resources can be performed to reduce the false positives brought by single-omics analysis. More importantly, the joint analysis of multi-omics data is more conducive to the study of phenotypic and biological process regulation mechanism of biological models. Integrated transcriptomic and metabolomic analysis refers to the normalization and statistical analysis of batch data from different levels of biomolecules, such as transcriptome and metabolome, to establish data relationships between different levels of molecules; at the same time, combined with functional analysis, metabolic pathway enrichment, molecular interactions and other biological functional analysis, to systematically and comprehensively analyze biomolecular functions and regulatory mechanisms, and finally achieve a comprehensive understanding of the general trends and directions of biological changes. At present, there are many ideas about the integration of transcriptional and metabolomic analysis, and the most common one is the integration of data based on the same KEGG pathway. Through comparative transcriptome analysis between different groups, we can find the differentially expressed genes involved in an important metabolic pathway. Meanwhile, in the results of metabolomics analysis, we focus on the relationship of metabolite changes in this pathway, and further explore which key genes are responsible for the metabolite changes, and analyze from the perspective of cause to effect. Thank you again for your valuable comments, and I hope that the improved manuscript will meet your expectations.

Point 5:Authors should carefully review the manuscript text to avoid typos and syntax error

 Few examples: Line 30 – ‘Databas’ instead ‘Database’. 

Response 5: I apologize for the many typos and syntax errors in the manuscript due to my negligence, which made your review a bad experience. I have made changes in the revised manuscript and sent the revised manuscript to the English language editor at MDPI for touch-ups. I hope the revised manuscript will meet your expectations and thank you again for your review of the manuscript and your valuable comments!

Round 2

Reviewer 1 Report

Thank you for the response. The technical questions are adequately addressed. The experiments were properly done, and the study is novel and provides interesting data. Nevertheless, the robustness and generalizability are limited by the in vitro work in few cell lines.  

Reviewer 2 Report

Thanks for providing a point-by-point response to my concerns, I could see the manuscript has improved significantly from its original version.

I recommend the article in this present form for publication in pathogens.